# Current Understandings of Core Pathways for the Activation of Mammalian Primordial Follicles

**DOI:** 10.3390/cells10061491

**Published:** 2021-06-13

**Authors:** Yu Zhao, Haiwei Feng, Yihui Zhang, Jian V. Zhang, Xiaohui Wang, Dongteng Liu, Tianren Wang, Raymond H. W. Li, Ernest H. Y. Ng, William S. B. Yeung, Kenny A. Rodriguez-Wallberg, Kui Liu

**Affiliations:** 1Center for Energy Metabolism and Reproduction, Shenzhen Institute of Advanced Technology, Chinese Academy of Sciences, Shenzhen 518055, China; jian.zhang@siat.ac.cn; 2Shenzhen Key Laboratory of Fertility Regulation, Center of Assisted Reproduction and Embryology, The University of Hong Kong-Shenzhen Hospital, Haiyuan First Road 1, Shenzhen 518053, China; fhw1010@hku.hk (H.F.); yihuihku@connect.hku.hk (Y.Z.); wangxh@hku-szh.org (X.W.); liudt@hku.hk (D.L.); wangtr@hku-szh.org (T.W.); raymondli@hku.hk (R.H.W.L.); nghye@hku.hk (E.H.Y.N.); wsbyeung@hku.hk (W.S.B.Y.); kliugc@hku.hk (K.L.); 3Department of Obstetrics and Gynecology, Li Ka Shing Faculty of Medicine, The University of Hong Kong, Hong Kong 999077, China; 4Shenzhen Key Laboratory of Metabolic Health, Shenzhen 518055, China; 5Guangdong-Hong Kong Metabolism & Reproduction Joint Laboratory, Shenzhen 518053, China; 6Department of Oncology-Pathology, Karolinska Institutet, 14186 Stockholm, Sweden; kenny.rodriguez-wallberg@ki.se

**Keywords:** primordial follicles, PI3K/PTEN signaling, in vitro activation, oocyte, granulosa cells, fertilization

## Abstract

The mammalian ovary has two main functions—producing mature oocytes for fertilization and secreting hormones for maintaining the ovarian endocrine functions. Both functions are vital for female reproduction. Primordial follicles are composed of flattened pre-granulosa cells and a primary oocyte, and activation of primordial follicles is the first step in follicular development and is the key factor in determining the reproductive capacity of females. The recent identification of the phosphatidylinositol 3 kinase (PI3K)/phosphatase and tensin homolog deleted on chromosome 10 (PTEN) signaling pathway as the key controller for follicular activation has made the study of primordial follicle activation a hot research topic in the field of reproduction. This review systematically summarizes the roles of the PI3K/PTEN signaling pathway in primordial follicle activation and discusses how the pathway interacts with various other molecular networks to control follicular activation. Studies on the activation of primordial follicles have led to the development of methods for the in vitro activation of primordial follicles as a treatment for infertility in women with premature ovarian insufficiency or poor ovarian response, and these are also discussed along with some practical applications of our current knowledge of follicular activation.

## 1. Introduction: The Development of Primordial Follicles Is Essential for Female Reproduction

The mammalian ovary is a key component of the female reproductive system and contains follicles at various developmental stages [1]. In addition to its role in producing mature oocytes, the ovary is also responsible for synthesizing and secreting hormones that are vital to reproductive development and fertility [2,3,4].

Follicles are the functional units of ovaries, and in most mammalian species primordial follicles are formed before or shortly after birth [5]. A large number of primordial follicles progressively occupy more peripheral parts of the ovary to create the primordial follicle pool [6], and the primordial follicle pool is considered to be the only source of germ cells for fertilization [5]. Although this theory has been challenged over the past decade [7,8], the latest research by Zhang et al. supports the traditional view that no follicular renewal occurs in postnatal life in mice and humans [9,10].

Once the primordial follicle pool is formed, the primordial follicles have three possible developmental fates—to maintain a quiescent state as primordial follicles, to be activated and enter the growth phase, or to undergo death [11]. The balance between the dormancy, activation, and death of primordial follicles is considered to be the decisive factor in determining the length of female reproductive life [12,13]. In humans, the initial primordial follicle pool at birth contains approximately 500,000–1,000,000 follicles in total. To maintain the lengthy reproductive lifespan of female mammals, most of the primordial follicles remain in a dormant state in the ovaries. Eventually, more than 99% of the primordial follicles will die with age, and this is referred to as ovarian aging [14,15]. In mammals, the quantity and quality of the primordial follicle pool determine the length of reproductive life [16,17], and menopause occurs in human females when the number of primordial follicles drops below approximately 1000 [18].

## 2. Activation of Primordial Follicles

The activation of primordial follicles and subsequent follicular development is an irreversible process [2]. The characteristics of follicular activation include the transformation of granulosa cells from a flattened to a cuboidal shape and an increase in oocyte volume [19,20]. Recent studies tracing the development of follicles have shown that there are two distinct populations of primordial follicles in the mouse ovary. The first wave of primordial follicles in the medulla of the ovary begins to be activated once they are formed, whereas the primordial follicles in the ovarian cortex are gradually activated over the course of the female reproductive lifespan [21,22].

Only a small proportion of primordial follicles are recruited into the growing follicle pool [16,23], and the phosphatidylinositol 3 kinase (PI3K)/phosphatase and tensin homolog deleted on chromosome 10 (PTEN) signaling pathway in oocytes plays a critical role in primordial follicle activation [24]. Studies have shown that PI3K signaling might be the only pathway for oocyte activation [25]. Once the PI3K/PTEN signaling pathway is activated, it can convert phosphatidylinositol 4,5-bisphosphate (PIP2) into phosphatidylinositol 3,4,5-trisphosphate (PIP3), which activates Akt and phosphorylates downstream forkhead transcription factor Foxo3. Foxo3 shuttles from the nucleus into the cytoplasm, resulting in primordial follicle activation [11,26]. The mechanistic target of rapamycin complex 1 (mTORC1) signaling pathway is another crucial pathway that initiates primordial follicle activation in pre-granulosa cells by activating PI3K/PTEN signaling [27]. In this article, we elaborate on the important roles of these two signaling pathways in primordial follicle activation.

## 3. The PI3K/PTEN Signaling in Oocytes Governs the Primordial Follicle Activation

According to their structural characteristics and substrate specificity, PI3Ks are divided into three classes (I, II, and III) [28]. Among them, class I PI3Ks are also divided into two subfamilies, class IA and class IB. Class IA PI3Ks, which are the most well-studied PI3Ks, are lipid kinases that consist of a p85 regulatory subunit and a p110 catalytic subunit, and they phosphorylate the 3′-OH group on the inositol ring of inositol phospholipids [29,30,31]. The p110 catalytic subunit can convert PIP2 into PIP3 [11,32]. PIP3 recruits both phosphatidylinositol-dependent kinase 1 (PDK1) and the protein serine–threonine kinase Akt to the cell membrane, where PDK1 phosphorylates and activates Akt [29,33]. Akt subsequently phosphorylates a range of substrates, including the cell-cycle inhibitor p27 (also known as p27KIP1), glycogen synthase kinase 3, tuberous sclerosis 2 (TSC2), and the forkhead box transcription factors (FOXO) families [34,35,36,37]. The PI3K signaling network is thus important for cell proliferation, cell cycle entry, and cell survival [38,39].

As a negative regulator of PI3K function, PTEN reverts PIP3 back to PIP2 and thus suppresses PI3K signaling [40,41]. For many decades, much attention has been focused on the role of PTEN as a tumor suppressor in cancers [38,42], but its key role in ovarian primordial follicle activation was not elucidated until 2008. Studies from Diego Castrillion’s group and Kui Liu’s group showed that oocyte-specific knockout of the *Pten* gene results in over-activation of the entire pool of primordial follicles, and this is followed by the premature depletion of all the primordial follicles, thus revealing the important roles of PI3K and PTEN in follicular activation [24,26] (Figure 1). Subsequently, the role of PTEN in primordial follicle activation has been demonstrated in other species. Bovine ovary fragments exposed to vanadate-derived dipotassium bisperoxo (5-hydroxy-pyridine-2-carboxylic) oxovanadate (V) (bpV(HOpic)), an inhibitor of PTEN, showed a significantly higher proportion of growing follicles compared to control, but was associated with DNA damage and impaired DNA repair competence [43]. For pig and sheep ovarian tissues, bpV(HOpic) affects ovarian follicle development by promoting the initiation of follicle development [44,45]. These large animal models may be helpful to optimize the follicular activation methods for transfer to clinical application. In humans, inhibition of PTEN with bpV(HOpic) affects human ovarian follicle development by promoting the initiation of follicle growth to the secondary stage, but the survival of isolated secondary follicles was severely affected [46]. However, a short-term study showed bpV(HOpic) treatment enhances the activation of primordial follicles in both fresh and cryopreserved human ovarian cortex, and enlarges growing populations without inducing apoptosis in follicles [47].

The forkhead transcription factor Foxo3 is a substrate of Akt and is an important factor downstream of the PI3K/PTEN signaling pathway, and it plays a specific and essential role in follicular activation [48,49]. The PI3K signaling pathway controls primordial follicle activation through Foxo3. Foxo3 is imported into the nucleus during primordial follicle formation and then is exported to the cytoplasm upon activation. [41]. *Foxo3^–/–^* female mice present with an ovarian phenotype of global follicular activation leading to oocyte death, early follicle depletion, and infertility [50]. Using immunofluorescence techniques, we double stained the ovarian sections of PD8 mice with Foxo3 and DDX4. Consistent with previous reports, our results show that Foxo3 shuttles from the nucleus into the cytoplasm once the follicles are activated (Figure 2).

## 4. TSC–mTORC1 Signaling in Pre-Granulosa Cells Initiates Primordial Follicle Activation

mTORC1 is a serine/threonine kinase that regulates cell growth and metabolism by promoting various metabolic processes such as protein, lipid, and organelle biosynthesis [51]. In mammals, mTORC1 activates p70 S6 kinase 1 (S6K1) and ribosomal protein S6 (rpS6) and inactivates eukaryotic translation initiation factor 4E (4E-BPs) to promote protein synthesis, ribosomal biogenesis, and cell growth [52,53].

In cells, mTORC1 is negatively regulated by a heterodimer complex consisting of TSC1 and TSC2 [51], which are the products of the tumor suppressor genes *Tsc1* and *Tsc2* [54]. The TSC1/TSC2 complex inhibits the activation of mTORC1 through the GTPase-activating protein domain of TSC2, and TSC1 maintains the stability of TSC2 by preventing its ubiquitination and degradation [55]. Some studies have shown that conditional knockout of *Tsc1* or *Tsc2* in mouse oocytes leads to the global activation of primordial follicles around the time of puberty, thus resulting in follicular depletion in early adulthood [56,57]. Moreover, deletion of *Tsc1* in pre-granulosa cells also results in overall primordial follicle activation [27], further suggesting that the TSC1/TSC2 complex plays an important role in regulating primordial follicular activation.

Another study from Liu’s group showed that conditional knockout of *Rptor* (mTOR regulatory protein complex 1) in pre-granulosa cells can inactivate mTORC1 signaling, and this suppresses follicular activation and prevents the awakening of dormant oocytes [27,58]. However, the inactivation of mTORC1 activity by deletion of *Rptor* in oocytes does not affect follicular development or female reproductive capacity [59]. These results suggest that mTORC1 signaling in pre-granulosa cells is essential for the activation of primordial follicles.

The first stage in primordial follicle activation is the differentiation and proliferation of pre-granulosa cells, which is followed by oocyte growth in the ovary [60,61,62]. Zhang et al. demonstrated that mTORC1 signaling in pre-granulosa cells activates Kit receptor on the oocyte surface by enhancing the expression of KIT ligand (KITL) in pre-granulosa cells from Tsc*1^−/−^* mouse ovaries [27]. They then crossed *Tsc1^−/−^* mice with *Kit^Y719F^/Kit^Y719F^* mice to further determine how the KITL–KIT and the intraoocyte PI3K signaling pathways function to awaken dormant oocytes. Phosphorylation of KIT Y719 is responsible for activating the downstream PI3K signaling, and *Kit^Y719F^/Kit^Y719F^* mice carry a point mutation that prevents this phosphorylation [27]. The Y719F mutation in the pre-granulosa cells of *Tsc1^–/–^*, *Kit^Y719F^/Kit^Y719F^* mice prevented the KIT-mediated activation of PI3K in oocytes. Thus, binding of KITL to Kit receptor results in the phosphorylation of KIT Y719, and this further activates PI3K signaling in mouse oocytes. The activated PI3K phosphorylates Akt, and this leads to the shuttling of Foxo3 from the nucleus to the cytoplasm resulting in the activation of primordial follicles [27] (Figure 3).

## 5. Other Related Pathways That Regulate Follicular Activation

### 5.1. Hippo Signaling and the Yes-Associated Protein (YAP) Pathway

In mouse models, the Hippo signaling pathway has been shown to play a role in regulating cell proliferation and apoptosis in order to maintain optimal organ size through growth inhibition [63,64]. Through a complex kinase cascade, the Hippo signaling pathway phosphorylates and inactivates the transcriptional coactivator YAP and transcriptional coactivator with PDZ-binding motif (TAZ) in order to inhibit organ growth [65]. Recent studies have shown that fragmentation of murine ovaries promotes actin dynamics and disrupts the Hippo signaling pathway, leading to the production of CCN (cellular communication network factor) and anti-apoptotic BIRC (baculoviral inhibitors of apoptosis repeat containing), the promotion of follicle growth, and the generation of mature oocytes [66]. Another recent study showed that mechanical signaling caused by internal or external forces can affect follicular activation through the Hippo and Akt pathways involving YAP, TAZ, PTEN, mTOR, and Foxo3 [2]. Based on the above findings, ovarian fragmentation via disruption of Hippo signaling followed by Akt stimulation shows promise as a new approach to promote follicular activation and growth for the treatment of infertility in patients with premature ovarian insufficiency (POI) [66,67].

### 5.2. Mitogen-Activated Protein Kinase (MAPK3/1) Pathways

MAPK is expressed in several mammalian tissues and plays important functions in regulating cell proliferation and differentiation, stress responses, and immune reactions [68,69,70]. MAPK3/1 activity has been studied in female mice for many years, and disruption of MAPK3/1 in mouse ovarian granulosa cells showed that these kinases are necessary for luteinizing hormone-induced oocyte resumption of meiosis, ovulation, and luteinization [71]. Studies in rat ovaries have shown that treatment with PD98059, a MAPK inhibitor, significantly inhibits activation of the primordial follicles, suggesting that MAPK signaling participates in primordial follicle activation and growth [72,73]. A recent study demonstrated that inhibition of MAPK3/1 signaling in mouse ovaries with the MAPK3/1 inhibitor U0126 decreases the number of growing follicles, the phosphorylation levels of Tsc2, S6K1, and rpS6, and the expression of KITL, indicating that MAPK3/1 signaling is involved in primordial follicle activation through mTORC1–KITL signaling in granulosa cells. Moreover, U0126 decreases the phosphorylation levels of Akt, suggesting that MAPK3/1 signaling regulates the activation of primordial follicles through the PI3K signaling in oocytes [74]. Therefore, MAPK3/1 activity plays an important role in primordial follicle activation through the mTORC1–KITL signaling pathway in pre-granulosa cells and through KIT–PI3K signaling in oocytes [74].

### 5.3. The 5′-Adenosine Monophosphate-Activated Protein Kinase (AMPK) Pathway

AMPK is a highly conserved serine–threonine protein kinase and is a key part of a kinase-signaling cascade that senses cellular energy status. In order to maintain energy homeostasis, AMPK regulates energy consuming and generating metabolic pathways [75]. AMPK activators reduce YAP-dependent CCN2 expression under conditions of energy stress, whereas AMPK depletion blocks this effect [76], and the increase in CCN2 expression results in ovarian follicle growth due to disruption of the Hippo signaling pathway [66]. In contrast, treatment with AMPK inhibitors results in activation of the mTOR signaling pathway, increased CCN2 expression, stimulation of follicular development, and promotion of ovarian angiogenesis [77]. Therefore, AMPK inhibition promotes follicular development through the AMPK–mTOR and AMPK–CCN2 pathways in mouse ovaries.

### 5.4. Cell Division Cycle 42 (CDC42)

CDC42 is a member of the Rho GTPase family and serves as a key regulator in controlling numerous cellular functions [78], and a recent study showed that CDC42 plays an important role in the activation of primordial follicles. The study showed that inhibition of CDC42 with a specific inhibitor significantly suppressed primordial follicle activation in mouse ovaries, while overexpression of CDC42 accelerated follicle activation [79]. CDC42 appears to function in the activation of primordial follicles by binding to the p110β catalytic subunit of PI3K and regulating PTEN expression in oocytes [79].

Recently, a study showed that treatment with epidermal growth factor (EGF) stimulates the activation of primordial follicles in both mouse ovaries and human cortical pieces using a new in vivo system (noninvasive ovarian topical administration) with dissolved EGF in liquid Matrigel, suggesting the role of EGF in follicle development [80]. Further evidence demonstrated that EGF promotes the activation of primordial follicles by increasing CDC42–PI3K signaling activity in mice. Combined with an inducible POI mouse model, that paper showed the effects of EGF in living animals and suggested new ideas for the treatment of female infertility [80].

### 5.5. Anti-Müllerian Hormone (AMH)

AMH is a dimeric glycoprotein belonging to the transforming growth factor-β superfamily of growth and differentiation factors [81] and was first identified as the causative factor behind the regression of the Müllerian ducts during fetal development in males [82,83]. AMH is not expressed in female rodents before birth [84,85], but it is readily detected in the granulosa cells of growing follicles a few days after birth [86,87]. In humans, AMH is produced by the granulosa cells from about 36 weeks post conception (wpc) until menopause [88,89]. Moreover, AMH also plays a role in primordial follicle activation. AMH treatment decreases the primordial to primary follicle transition in rat ovaries by inhibiting endogenous growth factors, such as KITL [90]. Studies in an AMH-*null* mouse model showed that a lack of *Amh* does not affect fertility in female mice [91], but primordial follicles are depleted earlier than those in the wild-type mice. Moreover, the ovaries of AMH-*null* female mice contain fewer primordial follicles and more growing follicles than those of wild-type mice [92]. Similar roles were identified in human ovarian cortical tissue biopsy specimens, suggesting that AMH inhibits the initiation of primordial follicle growth in vitro [93]. Taken together, these results suggest that AMH serves as a “brake” on primordial follicle activation and prevents the premature depletion of the primordial follicle pool. However, the mechanism through which AMH regulates the activation of primordial follicles is still not clear, and whether AMH plays a role in signal transduction by regulating PI3K signaling remains to be studied.

AMH is widely applied in the clinic for predicting the ovarian response to controlled ovarian stimulation treatment, because AMH is mainly produced in the growing follicles, which are capable of responding to exogenous gonadotrophins [94,95]. For in vitro fertilization (IVF) treatment, AMH is a good predictor of poor ovarian response, although its cut-off value varies depending on the assay that is used [96].

### 5.6. Insulin-Like Growth Factor (IGF) and Transforming Growth Factor Beta 1 (TGFβ1)

The IGF system, which is composed of insulin and insulin-like growth factors 1 (IGF-1) and 2 (IGF-2), participates in cell growth, proliferation, and survival [97]. IGFs, particularly IGF-1, can work synergistically with gonadotropins to stimulate follicular steroidogenesis [98]. IGF-1, either alone or with FSH, can increase AKT phosphorylation in granulosa cells [99]. The Akt and MAPK3/1 pathways are involved in mediating the effects of gonadotropins and IGF on follicle cell proliferation and follicular development [100], and IGF-1 can promote primordial follicle growth via the PI3K/AKT signaling pathway [101]. In addition, several studies have shown that IGF-1 crosstalks with the TGFβ family at multiple levels in cell lines [102], although further studies are needed to explore the relationship between these two essential pathways in ovary. As a consequence of such a central role, the Akt and MAPK3/1 pathways are the key signaling pathways that mediate the effect of IGF on regulating the development of follicles.

TGFβ1 is a member of the TGFβ superfamily of proteins, which function in the regulation of cell growth and differentiation [103,104]. Recently, TGFβ1 has been found to be involved in the maintenance of primordial follicles in mice [20]. The in vitro culture of mouse ovaries with TGFβ1 significantly inhibits the activation of primordial follicles, and TGFβ1 significantly induces oocyte apoptosis and inhibits somatic cell proliferation through regulation of the TSC/mTORC1 signaling pathway [20].

### 5.7. Liver Kinase B1 (LKB1)

LKB1 is a serine/threonine kinase that plays roles in regulating cellular energy homeostasis, the establishment of cell polarity, vascular development, and tumor suppression [105]. LKB1 regulates energy balance by phosphorylating AMPK at threonine 172 on the α subunit, and its function is critical for female fertility [106]. Germ line mutations in *LKB1* are associated with Peutz–Jeghers syndrome (PJS) in humans [107], and conditional knockout of *Lkb1* in mouse oocytes leads to excessive activation of the primordial follicle pool resulting in POI in adulthood [108]. Rapamycin can block follicular depletion in *Lkb1* conditional knockout mice, indicating that elevated mTOR signaling is responsible for the excessive follicle activation and growth [108]. Together, these results suggest that LKB1 is indispensable for maintaining the reserve of primordial follicles.

### 5.8. The Cyclin-Dependent Kinase Inhibitor 1B (p27)

p27 is a regulatory protein that controls cell cycle progression by binding to cycle-dependent kinase complexes [109,110]. It is a negative regulator of cell cycle progression and is a key molecule in mammalian ovarian development [111]. The role of p27 as a tumor suppressor has been widely studied in human cancer [112]. p27 is expressed in the nuclei of dormant oocytes in mouse ovaries [113] and is considered to be a substrate of PI3K–Akt signaling, which regulates cell survival and cell cycle entry [11,35]. Studies in *p27*-deficient mice have shown that follicle assembly is accelerated and that the primordial follicle pool is prematurely activated, leading to POI in early adulthood [113]. Overall, p27 plays an important role in maintaining the primordial follicle pool and preventing follicles from being activated prematurely.

### 5.9. Semaphorin 6C (Sema6c) and Sirtuin 1 (SIRT1)

Recent studies of semaphorins have focused on their roles in cancer, the immune system, the vascular system, and organogenesis, and Sema6c has been found to play a role in the neonatal mouse ovary [114,115,116,117,118]. The latest research has shown that downregulation of SEMA6C results in the activation of primordial follicles through interactions with the PI3K–AKT–rpS6 signaling pathway, indicating that Sema6c is involved in primordial follicle dormancy [119]. These results give us valuable information for understanding POI and ovarian aging.

The sirtuins are a family of NAD^+^-dependent deacetylases that have recently emerged as key sensors in metabolism, cancer development, and aging [120,121]. As the most conserved sirtuin, Sirt1 has been shown to play crucial roles in mammalian health and disease [122]. The expression of SIRT1 in oocyte nuclei has been shown to increase during primordial follicle activation [123], and the function of Sirt1 is dependent on some transcription factors, including Foxo3 [124]. Based on these results, it can be hypothesized that SIRT1 is an important factor in determining the fate of the primordial follicles downstream of the PI3K–AKT–Foxo3a signaling pathway.

### 5.10. Exosomes

Exosomes derived from mesenchymal stem cells (MSC-exos) have great therapeutic potential due to their ethical acceptance, abundant tissue sources, low immunogenicity, and rapid renewal properties [125]. Like their cellular sources, MSC-exos can help in organizing and maintaining a stable environment for tissues and can restore critical cellular functions by initiating the process of repair and regeneration [126]. Among the components of MSC-exos, miRNAs are considered to be vital factors for regulating target cellular functions via several key signaling pathways that are also related to primordial follicle activation, including activation of the PI3k–Akt and mTOR pathways via miR29, miR24, miR34, miR130, and miR378 in cardioprotection [127], regulation of PTEN expression via miR-17–92 clustering in neurological recovery [128], and attenuation of TGF-β receptor 1 expression via miR-let7 in renal fibrosis [129].

With regards to female fertility, an increasing number of studies have shown that MSCs can improve ovarian function in POI or natural-aging animal models [130,131,132]. Moreover, transplantation of human umbilical cord mesenchymal stem cells (HucMSCs) has also been applied in the clinic as a therapy for POI patients [133]. However, the potential molecular mechanism of HucMSCs in treating POI remains unclear. Recently, exosomes derived from human umbilical cord mesenchymal stem cells (HucMSC-exos) have been shown to promote follicular activation and development in newborn mouse ovaries and to delay the age-related retardation of fertility in female mice [134]. It has also been shown that HucMSC-exos stimulate primordial follicle activation through the PI3K/mTOR signaling pathway via their miRNA components, such as miR-146a-5p and miR-21-5p [134]. Thus, HucMSC-exos might represent a new treatment method for women with diminished ovarian reserve to enhance their decreased fertility.

## 6. Application of In Vitro Activation of Primordial Follicles in the Treatment of Poor Ovarian Response (POR) and POI

In recent years, more and more women in many societies are delaying marriage and postponing motherhood, and this is increasing the number of women seeking in vitro fertilization (IVF) in order to have their own children. However, IVF cannot help women with POR or POI because they only have undeveloped follicles in their ovaries, and these do not respond to hormones. In recent years, primordial follicle in vitro activation (IVA) has provided a new potential treatment for infertile women with POR and POI. IVA is a new in vitro assisted reproductive technology that treats infertility in POR patients by activating the primordial/small follicles in vitro [135]. Based on studies of the PTEN–PI3K molecular network and the Hippo signaling pathway in the activation of primordial follicles, IVA technology has brought hope for the treatment of infertility in POR patients [66,136].

In 2013, Dr. Aaron Hsueh [66] disrupted the Hippo signaling pathway by fragmenting ovaries followed by Akt stimulation for treating infertility in POI patients. First, the ovaries were removed by laparoscopic surgery and cortical strips were obtained. Histological analyses were performed to detect residual follicles using a small portion of the ovarian cortices, and the rest of the cortical strips were then vitrified and cryopreserved. Before transplantation, the strips were thawed and further cut into 1–2 mm^3^ cubes followed by treatment with Akt stimulators for two days. The cubes were then autografted beneath the serosa of Fallopian tubes for development into follicles. After antral follicles had appeared, the patients were injected with FSH. With the emergence of preovulatory follicles, hCG was injected for the final maturation of the oocytes. These oocytes were aspirated and used in IVF to form embryos (Figure 4). In their paper, a total of 27 patients participated in the IVA treatment, and 13 of them were found to have follicles in their ovaries by histological analyses. Using IVA methods, the follicles were able to grow and reach the pre-ovulation stage in eight patients (62%). Of these eight patients, mature oocytes were successfully obtained from five patients for intracytoplasmic sperm injection. Three of the patients each successfully obtained two embryos, and at the time the article was published one of the patients had a detected pregnancy and one other had successfully delivered a healthy baby.

Similar results were also observed in a study of IVA including 37 POI patients. Among these patients, 54% still had residual follicles in their ovaries, 9 out of 20 (45%) showed follicular development, and oocytes were successfully obtained from the growing follicles of six patients. Following IVF, embryo transfers were carried out in four patients and three pregnancies were identified that resulted in one miscarriage and two successful deliveries [137]. In 2016, an article reported a simplified IVA protocol for transplanting fresh tissues instead of frozen-thawed tissues. In that paper, 6 out of 14 (43%) patients showed follicular development and four patients had successful oocyte retrieval of six oocytes. Four early embryos were observed after IVF, and after the embryo transfer one patient delivered a healthy boy [138].

In recent years, drug-free IVA, i.e., only fragmentation of the ovarian cortex, has appeared as a simplified IVA procedure for POR and POI patients and has been reported to allow retrieval of more oocytes and to achieve higher pregnancy rates. Drug-free IVA showed that disrupting the Hippo signaling pathway through ovarian cortical fragmentation could effectively promote follicle growth and increase the clinical pregnancy rate. Using this simplified drug-free IVA procedure, two case reports showed successful pregnancies in patients with POI in 2018 and 2019 [139,140]. Another study used a similar drug-free IVA method to treat 20 women with diminished ovarian reserve (DOR). Patients were randomized to have four biopsies taken from one ovary by laparoscopy, followed by fragmentation of the ovarian cortex, then auto transplanted to a peritoneal pocket. The contralateral ovary served as a control. Growth of four follicles was detected in three patients (15%), and one oocyte was retrieved and fertilized, but embryonic development failed [141]. In another paper, follicular growth waves were observed in 9 out of 11 patients (81%) treated with this simplified IVA procedure. One patient showed a natural pregnancy, and 16 embryo transfers in five patients resulted in one live birth, two ongoing pregnancies, and one miscarriage [142]. The paper published by Janisse et al. reported that 7 out of 14 (50%) patients showed follicular development, and five patients had successful retrieval of seven oocytes. Six embryo transfers were carried out in five patients resulting in four pregnancies [143]. Together, these results suggest that the drug-free IVA procedure may be efficient in treating some patients with POI, POR, or DOR. At the 2019 Summit Meeting of Ovarian Disease and Reproductive Health (Shenzhen, China), Dr. Aaron Hsueh showed the effectiveness of drug-free IVA in the treatment of POR and POI patients. This new approach only involves mechanical cutting of the ovarian cortex and does not require tissue culture, and thus only a single surgery is required.

Research into mTORC1 signaling in the activation of primordial follicles has shown that the efficiency of primordial follicle activation can be improved by the combination of mTOR stimulators and PI3K activators [144]. This suggests that combinations of drugs affecting multiple signaling pathways might have a better effect on the activation of primordial follicles. Recently, a review investigated the mechanism of DNA damage surveillance and its association with the PI3K/PTEN signaling pathway, as well as its influence on the DNA damage response (DDR) during primordial follicle activation and ovarian aging. Focusing on the mechanism of DDR in oocytes may provide new approaches to improve the clinical regulation of primordial follicular activation [145].

Although IVA has been used in the treatment of POI patients, there is still much to do to improve this approach. Due to the complex clinical manifestations, complex etiology, and heterogeneity of POI, the application of this technology needs to consider individual differences among patients. The use of drugs for activating primordial follicles might result in negative feedback from downstream effectors. For example, inhibition of PTEN will not always leads to a constant over-activation of the PI3K pathway, but rather the opposite, which might lead to unsatisfactory clinical outcomes and low success rates of IVA.

## 7. Conclusions

In this review, we highlight the roles of the PI3K/PTEN signaling pathway in primordial follicle activation along with other pathways and factors involved in the regulation of dormancy and the activation of primordial follicles (Figure 5). Primordial follicle activation needs to be further studied because a better understanding of this process will not only help to reveal the molecular mechanisms underlying follicular development, but will also facilitate the clinical treatment of infertility, including POR patients. At the same time, more target molecules need to be identified in order to develop more personalized treatments for individual patients.

## Figures and Tables

**Figure 1 cells-10-01491-f001:**
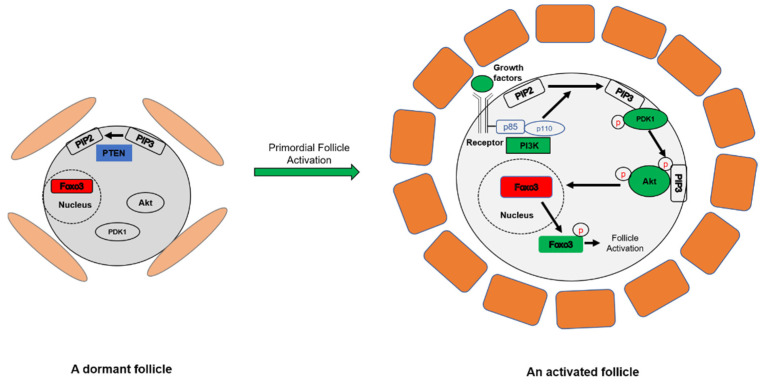
PI3K/PTEN signaling controls the activation of primordial follicles in the ovary. In dormant oocytes (left), PTEN reverts PIP2 back to PIP3 and prevents follicular activation. The downstream transcription factor Foxo3 is expressed in the nucleus. In growing oocytes (right), the PI3K–Akt signaling pathway is activated by converting PIP2 to PIP3. PIP3 then recruits both PDK1 and Akt to the cell membrane where PDK1 phosphorylates and activates Akt, resulting in Foxo3 hyperphosphorylation and nuclear export, which triggers the activation of the primordial follicle.

**Figure 2 cells-10-01491-f002:**
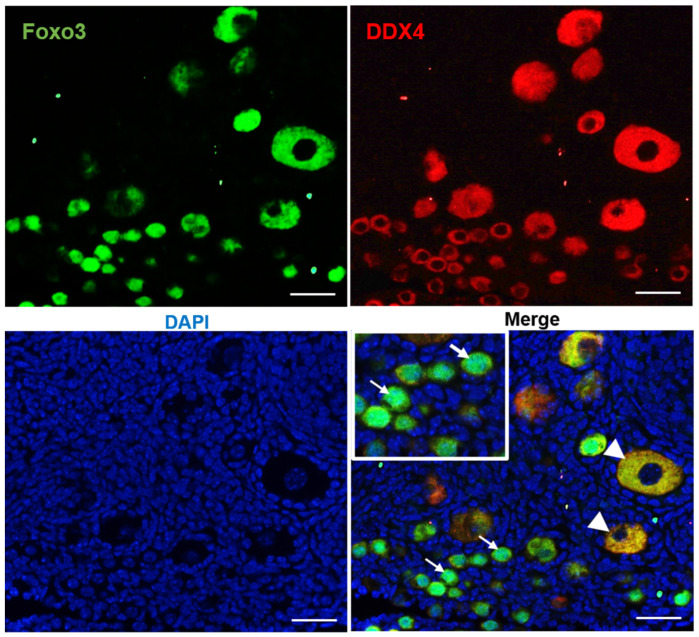
The expression of Foxo3 in primordial follicles and growing follicles. Foxo3 (green) is expressed in the nucleus of primordial follicles. In growing follicles, Foxo3 is translocated to the cytoplasm. DDX4 (red) was used as the germ cell marker, and DAPI (blue) was used to stain DNA. Arrowheads: oocyte cytoplasm with Foxo3. Arrows: oocyte nuclei with Foxo3. Scale bars: 50 µm.

**Figure 3 cells-10-01491-f003:**
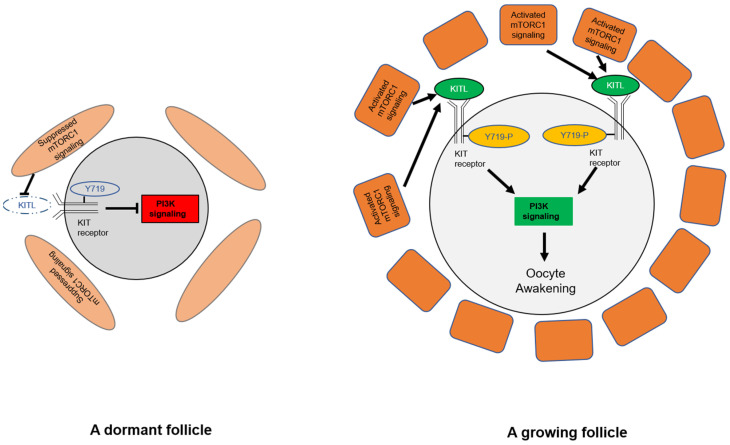
Pre-granulosa cells initiate the activation of primordial follicles through mTORC1–KITL in pre-granulosa cells and KIT Y719–PI3K in oocytes. In dormant follicles (**left**), the mTORC1 signaling in pre-granulosa cells is suppressed, and only a low level of KITL is expressed by pre-granulosa cells, which is insufficient to activate KIT Y719 on oocytes. KIT Y719 is not phosphorylated and the intraoocyte PI3K signaling is suppressed in this situation. In growing follicles (**right**), activated mTORC1 signaling in the pre-granulosa cells leads to enhanced production of KITL that binds to c-kit and subsequently activates the intraoocyte PI3K signaling through phosphorylation of KIT Y719. Activated PI3K signaling in oocytes awakens the dormant oocytes.

**Figure 4 cells-10-01491-f004:**
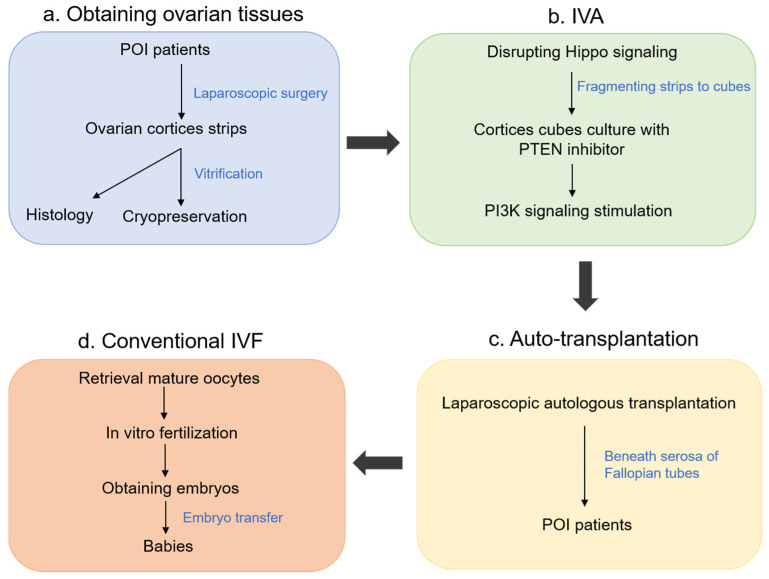
Schematic diagram of IVA. (**a**) Obtaining ovarian tissues. Under laparoscopic surgery, ovaries were obtained and cut into strips for histology and cryopreservation after vitrification. (**b**) IVA. After thawing of cryopreserved ovarian strips, the strips were fragmented into 1–2 mm^3^ cubes and cultured with Akt stimulators for 2 days. (**c**) Auto-transplantation. After two days of culture, the ovarian cubes were autografted beneath the serosa of the fallopian tubes. (**d**) Conventional IVF. Follicle growth was stimulated by injection of FSH. When preovulatory follicles were found, mature oocytes were retrieved for IVF to obtain embryos.

**Figure 5 cells-10-01491-f005:**
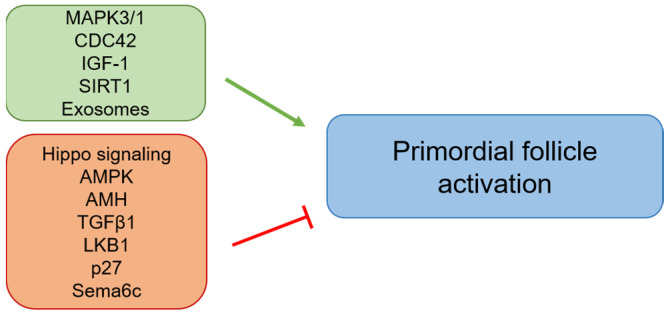
The roles of other pathways and molecules in primordial follicle activation. The green arrow indicates activation of primordial follicles, and the red arrow indicates inhibition of primordial follicle activation.

## Data Availability

No new data were created or analyzed in this study. Data sharing is not applicable to this article.

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
