# Peer review of "Current Understandings of Core Pathways for the Activation of Mammalian Primordial Follicles"

_cells, 2021, doi:10.3390/cells10061491_

Round 1

Reviewer 1 Report

This manuscript intended for a review article at Cells has summarized and updated the knowledge on the molecular mechanisms leading to activation of primordial follicles, a biological process that is critical for successful reproduction in mammals including human. The description of the key concept and the recent research advancement seems to be comprehensive and straightforward enough, and is accompanied by a wealthy reference list. In my view, the addressed topic might be of keen interest to the general readership of Cells, but the manuscript may require some improvement on the following points.

  1. The title of the manuscript somehow sounds like a headline of an experimental research article rather than a review. It delivers a highly focused and committed message that PI3K/PTEN signaling serves as the core mechanism in controlling primordial follicle activation. This concept is more intensified in the early part of the manuscript where it was stated, “PI3K signaling might be the only pathway for oocyte activation (lines 68-69).” The following sections describe the supporting evidence for the idea. However, the section 5 and its subsections are actually about “other related pathways that regulate follicular pathways,” and this latter part of the manuscript does not really show clear relevance to the PI3K/PTEN pathway. The section 5 is also very long with numerous subsections, and overall its contribution to the manuscript nearly outweighs the first part that is directly related to the PI3K/PTEN pathway. From the title and introduction, the readers expect the whole manuscript would revolve around this concept, but the later part of the manuscript only dilutes the message of PI3K/PTEN and causes confusion. Hence, the first thing for the authors to consider is to change the title or the structure of the text in a way that the entire manuscript matches the title.

  1. Figure 1 and Figure 3 are quite similar to each other. Perhaps they explain different aspects or pathways of primordial follicle activation, but their hierarchical relationship is not clear. This point should be explained more efficiently. Related to that, some of the “other related pathways” featured in section 5 were mentioned to have a link to the PI3K/PTEN pathway. However, they are all scattered, and it is not clear how such findings would fit into the theme illustrated in Fig. 1 or 3. In my view, it will be useful for the readers to have a ‘composite summary figure’ that summarizes all these relevant findings in one schematic picture.

  1. Figure 2 appears to be a piece of experimental data that has not been published before, but the explanation provided in the text or in the legend is far from sufficient. Once the data were provided, the related discussion should be more thorough: i) what is the difference between the two classes of oocytes in the same visual field, and what is its significance?, ii) how come the small oocytes in the inset totally lack cytoplasm?, iii) what are the experimental conditions, and how foxo3 and DDX4 were visualized? Are they immunofluorescence or the signals from red and green fusion proteins? iv) How would the same signals appear in the foxo3-/- mouse. If the data are available, it would be better to be exhibited here. This figure simply invites a series of question, and should thus be elaborated more to strengthen the point that the authors like to make.

  1. Section 5.10 about cdc42 is redundant to section 5.4 that bears the same title. It appears that section 5.10 is a truncated remnant of section 5.4. If it is so, the whole paragraph should be deleted and the subsection numbers rectified.

  1. For the reasons specified above, the manuscript somehow gives the impression that it is rather a summary of the multiple pathways leading to primordial follicle activation. Many of them can be grouped together under the umbrella of PI3K/PTEN pathway, while many others apparently cannot. Hence, overall, the manuscript was presented as an updated summary on the given topic rather than as a ‘novel perspective’ that really features PI3K/PTEN as a central counteracting mechanism that converges all the signals to act on the dynamic PIP2/PIP3 shifts in order to control the primordial follicle activation. It is a pity that this shortfall undermines the clinical implication and other intriguing merits of the manuscript.                    

Author Response

This manuscript intended for a review article at Cells has summarized and updated the knowledge on the molecular mechanisms leading to activation of primordial follicles, a biological process that is critical for successful reproduction in mammals including human. The description of the key concept and the recent research advancement seems to be comprehensive and straightforward enough, and is accompanied by a wealthy reference list. In my view, the addressed topic might be of keen interest to the general readership of Cells, but the manuscript may require some improvement on the following points.

  1. Comment: The title of the manuscript somehow sounds like a headline of an experimental research article rather than a review. It delivers a highly focused and committed message that PI3K/PTEN signaling serves as the core mechanism in controlling primordial follicle activation. This concept is more intensified in the early part of the manuscript where it was stated, “PI3K signaling might be the only pathway for oocyte activation (lines 85-86).” The following sections describe the supporting evidence for the idea. However, the section 5 and its subsections are actually about “other related pathways that regulate follicular pathways,” and this latter part of the manuscript does not really show clear relevance to the PI3K/PTEN pathway. The section 5 is also very long with numerous subsections, and overall its contribution to the manuscript nearly outweighs the first part that is directly related to the PI3K/PTEN pathway. From the title and introduction, the readers expect the whole manuscript would revolve around this concept, but the later part of the manuscript only dilutes the message of PI3K/PTEN and causes confusion. Hence, the first thing for the authors to consider is to change the title or the structure of the text in a way that the entire manuscript matches the title.

Response: We have changed the title to “Current Understandings of Core Pathways for the Activation of Mammalian Primordial Follicles” and modified some contents of Part 5 to make the title match the content.

  1. Comment: Figure 1 and Figure 3 are quite similar to each other. Perhaps they explain different aspects or pathways of primordial follicle activation, but their hierarchical relationship is not clear. This point should be explained more efficiently. Related to that, some of the “other related pathways” featured in section 5 were mentioned to have a link to the PI3K/PTEN pathway. However, they are all scattered, and it is not clear how such findings would fit into the theme illustrated in Fig. 1 or 3. In my view, it will be useful for the readers to have a ‘composite summary figure’ that summarizes all these relevant findings in one schematic picture.

Response: We appreciate the reviewer’s suggestions. Figure 1 focuses on the PI3K signaling pathway in oocytes, while Figure 3 focuses on the mTORC1 signaling pathway in pre-granulosa cells. The mTORC1 signaling pathway in pre-granulosa cells initiates primordial follicle activation through PI3K signaling in oocytes. These two signaling pathways play a critical role in the activation of primordial follicles, and this is why we use two diagrams to illustrate these two pathways. As for other pathways and molecules, we have added Figure 5 to summarize these molecules related to primordial follicle activation.

  1. Comment: Figure 2 appears to be a piece of experimental data that has not been published before, but the explanation provided in the text or in the legend is far from sufficient. Once the data were provided, the related discussion should be more thorough: i) what is the difference between the two classes of oocytes in the same visual field, and what is its significance?, ii) how come the small oocytes in the inset totally lack cytoplasm?, iii) what are the experimental conditions, and how foxo3 and DDX4 were visualized? Are they immunofluorescence or the signals from red and green fusion proteins? iv) How would the same signals appear in the foxo3-/- mouse. If the data are available, it would be better to be exhibited here. This figure simply invites a series of question, and should thus be elaborated more to strengthen the point that the authors like to make.

Response: i) There are two types of follicles displayed in the same field of vision: primordial follicles and growing follicles. The growing follicles showed cuboidal granulosa cells and larger oocytes compared with the primordial follicles (flat granulosa cells and small oocytes). Foxo3 is expressed in the nucleus of primordial follicles, and once a primordial follicle is activated Foxo3 shuttles from the nucleus into the cytoplasm. Foxo3 can be used as a marker to distinguish primordial follicles from growing follicles, which is its important significance.

  1. ii) DDX4 is specifically expressed in the cytoplasm of germ cells, so it can be seen from the figure of DDX4 (red signal) that small follicles have cytoplasm. However, due to the strong Foxo3-positive signal in the nucleus of small follicle oocytes and the small follicles with less cytoplasm, the oocytes of small follicles appear green in the merged picture, but in the enlarged merged picture (white box) the cytoplasm of small follicles appears yellow (red overlapped on green).

iii) Using immunofluorescence techniques, we double-stained the ovarian sections of PD8 mice with Foxo3 (rabbit) and DDX4 (mouse) and incubated them with the Goat anti-rabbit 488 and Goat anti-mouse 594 secondary antibodies. In this way, Foxo3 will display a green signal and DDX4 will display a red signal. We have included details of the experimental conditions in the manuscript (Line 114-118).

  1. iv) As for the results of Fig.2, we double-stained Foxo3 and DDX4 in wild-type mice. If we use newborn Foxo3-/- mice, DDX4 signaling is not different from that of wild-type mice. If we use the knockout mice around PD14, these mice should have larger ovaries and early-growing follicles in their ovaries and therefore should show more DDX4 signaling. However, Foxo3-/- mice should not have a positive signal for Foxo3 antibodies in their ovaries.
  2. Comment: Section 5.10 about cdc42 is redundant to section 5.4 that bears the same title. It appears that section 5.10 is a truncated remnant of section 5.4. If it is so, the whole paragraph should be deleted and the subsection numbers rectified.

Response: We apologize for this mistake, and we have deleted the duplicate part.

  1. Comment: For the reasons specified above, the manuscript somehow gives the impression that it is rather a summary of the multiple pathways leading to primordial follicle activation. Many of them can be grouped together under the umbrella of PI3K/PTEN pathway, while many others apparently cannot. Hence, overall, the manuscript was presented as an updated summary on the given topic rather than as a ‘novel perspective’ that really features PI3K/PTEN as a central counteracting mechanism that converges all the signals to act on the dynamic PIP2/PIP3 shifts in order to control the primordial follicle activation. It is a pity that this shortfall undermines the clinical implication and other intriguing merits of the manuscript.                    

Response: We appreciate the reviewer’s suggestions. We have revised the title to highlight the important signaling pathways found in recent years for primordial follicle activation. At the same time, we also made some deletions and modifications to the content of the manuscript to emphasize the clinical implication of primordial follicle activation.

Reviewer 2 Report

The authors aimed to review and systematically summarizes the roles of the PI3K/PTEN signaling pathway in primordial follicle activation and discusses how the pathway interacts with various other molecular networks to control follicular activation.

The study covers some concerns that have been overlooked in other similar topics, but some  issues should be improved before publication. The manuscript needs moderate English change and grammar correction. Please also check typos thorough the text.

Conclusion Section: This paragraph is missing. Please add it (removing from discussion section), including add some "take-home message".

Reviewer 3 Report

The current submitted version of the article entitled: " PI3K/PTEN Signaling Serves as the Core Pathway for the Activation of Mammalian Primordial Follicles" is highlighting different factors and pathways involved in the regulation of dormancy and the activation of primordial follicles. This issue is important to be deeply explored with more understanding of unknown cellular and molecular mechanisms of primordial follicle activation especially to suggest some innovative therapeutic approaches for patients with POI or poor ovarian response.

However, there are some revisions to do to improve the quality of the manuscript.

  • The part of "activation of primordial follicles" needs to be more developed highlighting the process step by step in details (from the increase of intracellular of PIP3 which is determined by the balance PI3K/PTEN. Then, the trigger of PI3K/PTEN/AKT and TSC/mTOR pathways are critical to lead the activation of the primordial follicles involving FOXO3 phosphorylation or inactivation).
  • Then, the next part could be described more in detail but I suggest making the title shorter "The main discovery ... follicles".
  • The role of PTEN needs to be developed in this part since PI3K and PTEN are both critical for the activation process.
  • I suggest making the title shorter and more informative of the part "TSC-mTORC1... oocytes"
  • In the AMH part, add the experience of the treatment of poor responders with AMH as molecules to enrich the reflection
  • The part of CDC42 is repeated!
  • The Akt and ERK pathways are the key signaling pathways to mediate the IGF effect which is important with members of the TGFb family (GDF9, bMP15, activin, inhibin, AMH, ...) to regulate the development of follicles. However, it is not well cited in the manuscript.
  • Moreover, there is a need to add a part describing the role of the activation of KIT signaling pathways in oocytes through KITL.
  • The exosomes part is superficially discussed without mentioning the role of some miRNA to regulate the activation of primordial follicles. 
